# In Vitro and in Field Response of Different Fungicides against *Aspergillus flavus* and *Fusarium* Species Causing Ear Rot Disease of Maize

**DOI:** 10.3390/toxins11010011

**Published:** 2019-01-01

**Authors:** Mario Masiello, Stefania Somma, Veronica Ghionna, Antonio Francesco Logrieco, Antonio Moretti

**Affiliations:** Institute of Sciences of Food Production, Research National Council (ISPA-CNR), Via Amendola 122/O, 70126 Bari, Italy; stefania.somma@ispa.cnr.it (S.S.); veronica.ghionna@ispa.cnr.it (V.G.); antonio.logrieco@ispa.cnr.it (A.F.L.); antonio.moretti@ispa.cnr.it (A.M.)

**Keywords:** prothioconazole, boscalid, fludioxonil, SDHI-resistance, Demethylation Inhibitors, maize ear rot disease control

## Abstract

*Aspergillus flavus*, the main aflatoxin B_1_ producing fungal species, *Fusarium graminearum*, a deoxynivalenol producer, and the fumonisin-producing species *F. proliferatum* and *F. verticillioides* are the main toxigenic fungi (TF) that colonize maize. Several strategies are available to control TF and related mycotoxins, such as chemical control. However, there is poor knowledge on the efficacy of fungicides on maize plants since few molecules are registered. The sensitivity of *F. graminearum*, *F. proliferatum*, *F. verticillioides*, and *A. flavus* to eleven fungicides, selected based on their different modes of action, was evaluated in both in vitro assays and, after selection, in the field. In vitro, demethylation inhibitors (DMI) showed excellent performances, followed by thiophanate-methyl and folpet. Among the succinate dehydrogenase inhibitors (SDHI), isopyrazam showed a higher effectiveness against Fusarium species than boscalid, which was ineffective against Fusarium, like the phenyl-pyrrole fludioxonil. Furthermore, both SDHIs and fludioxonil were more active against *A. flavus* than Fusarium species. In field trials, prothioconazole and thiophanate-methyl were confirmed to be effective to reduce *F. graminearum* (52% and 48%) and *F. proliferatum* contamination (44% and 27%). On the other hand, prothioconazole and boscalid could reduce *A. flavus* contamination at values of 75% and 56%, respectively.

## 1. Introduction

Cereals represent the most important crops providing the main carbohydrate source in human and livestock diets. Among cereals, maize is the first cultivated cereal worldwide [1]. However, under specific pedoclimatic conditions, abiotic and biotic stresses could cause huge economic and productive losses to maize cultivation. Among the fungal genera, Ustilago, Fusarium, and Aspergillus can appear from flowering stage to harvest time and are mainly associated to ear diseases [2,3,4,5,6,7,8,9]. In particular, great concern is caused by the diseases caused by mycotoxigenic species such as those belonging to Fusarium and Aspergillus genera. These species can contaminate maize kernels at maturity and produce several mycotoxins that can accumulate in the final products [10,11]. The trichothecenes deoxynivalenol (DON) and nivalenol (NIV), both well-known inhibitors of protein synthesis [12], the polyketide-derived fumonisins (FBs), associated with several animal and human diseases, the hestrogenic compound zearalenone (ZEA), and the carcinogenic aflatoxins are the major toxins detected in maize products [13,14,15].

Among maize crop diseases, the so called “Fusarium maize ear rot” (FER), caused by a complex of Fusarium species (e.g., *Fusarium graminearum* Schwäbe, *Fusarium proliferatum* (Matsushima) Nirenberg, and *Fusarium verticillioides* (Saccardo) Nirenberg), is very common in all maize cultivated areas [11]. Furthermore, Aspergillus species, occurring on maize kernels both in preharvest and during storage, also represent a serious problem since they cause high productive losses. Finally, since they can accumulate their mycotoxins in colonized tissues, and these mycotoxins are stable during food and feed processing procedure, these diseases have a significant economic impact and pose a serious risk for animal and human health [16,17,18,19].

Fusarium species can colonize maize in a wide range of climatic areas, and can alternate in the colonization of maize plants. In particular, while *F. graminearum*, the highest producing species of DON, NIV, and ZEA, better adapts to temperate geographic areas; *F. proliferatum* and *F. verticillioides*, representing the most important FBs producers [12], colonize maize grown in geographical areas with higher temperatures.

Fumonisins are mycotoxins often detected in association with fusaric acid, moniliformin, beauvericin, and fusaproliferin [10,20]. In central Europe, *F. verticillioides* is isolated mainly in co-occurrence with *F. subglutinans* and *F. temperatum*, while in southern Europe the spread of *F. verticillioides* is reinforced by the widespread presence of *F. proliferatum*, capable of producing FB_1_, moniliformin, beauvericin, and fusaproliferin [10].

*Aspergillus flavus*, the most important worldwide species able to produce aflatoxins on maize, colonizes this crop mostly when water stress and high temperature occur. *Aspergillus flavus* is commonly spread in tropical areas, however, due to climatic changes, this species is considered an emerging problem in several areas of Europe [21].

There is an urgent need to control these mycotoxigenic fungi on maize since the concern caused by their occurrence is growing worldwide [19,22].

According to the new strategies of integrated pest management (IPM), several agronomic, genetic, biological techniques, and agricultural practices are now available to prevent or limit fungal diseases and related mycotoxin accumulation. The use of new hybrids and the control of the European corn borer can contribute to reduce Fusarium and FB contamination [23,24]. Also, the sowing time was shown to have a significant impact in the reduction of FB contamination [25]. In recent decades, several researches, encouraged by European Union, focused on biological control strategies effective in reducing mycotoxigenic fungi in the field and their mycotoxins [26]. However, chemical control is still considered a key tool to limit the fungal diseases on many important food crops, such as the small cereals, where, for the direct control of Fusarium Head Blight (FHB), spraying fungicides at flowering has been included among good agricultural practices [27].

Nowadays, the chemical control of fungal pathogens can be achieved by several different target site fungicides, discriminable for their mode of action. According to IPM strategies schedules, fungicides with different modes of action can be used in mixture or in an alternating regime on the same crop. The most recent target site fungicides are the succinate dehydrogenase inhibitors (SDHIs), in addition to the well-known phenyl-pyrroles (PP fungicides), that affect the fungal osmotic signal transduction cascade and pathogen osmoregulation (fludioxonil is the most known “compound”), and to the benzimidazole carbamates and the demethylation inhibitors fungicides (DMIs), that affect sterol biosynthesis in membranes. 

DMIs and PPs are considered the most effective molecules registered nowadays to control fungal diseases caused by ascomycetes fungi, and also thought to be reliable on some cereal crops, where they are registered.

However, poor knowledge on the efficacy of fungicides to control fungal and mycotoxins contamination on maize, is available [28]. Furthermore, the occurrence of field isolates resistant to fungicides [29,30,31,32] and the need to reduce chemical residues on the final products at harvest could make it more difficult the management of fungal diseases.

The present work aimed to (a) evaluate the in vitro sensitivity of *F. graminearum*, *F. proliferatum*, *F. verticillioides*, and *A. flavus*, the most toxigenic fungal species occurring on maize, towards 11 fungicides currently used on several crops, since they are representative of different modes of action and, based on in vitro results, (b) test the most effective fungicides on maize plants grown in field.

Our final goals were (a) to improve the knowledge about the effectiveness of the most important molecules available on market, to control main mycotoxigenic fungi involved in maize ear rot and (b) to obtain useful information for the registration of chemical compounds on maize crop.

## 2. Results

### 2.1. In Vitro Colony Growth Inhibition

The mycelia growth inhibition was evaluated at 3, 5, 7, and 10 days of incubation. The mean values of each fungal species, based on three strains for each species, at 10 days of incubation are reported in Figure 1. The total data of the experiment are reported in Appendix A.

#### 2.1.1. DMIs

All DMIs showed an excellent activity to inhibit mycelial growth, up to 10 days of incubation. 

In particular, prothioconazole and prochloraz completely inhibited the colony growth of all fungal strains at the three concentrations tested, as shown in Figure 1. 

##### Metconazole

All Fusarium and *A. flavus* strains tested were inhibited by metconazole at the two highest doses (90 and 9 mg L^−1^); with the exception of *A. flavus* ITEM 8095 strain (inhibition value of 90% after 10 days of incubation, Appendix A). However, at the lowest concentration (0.9 mg L^−1^), a different response to metconazole was observed among fungal genera (Figure 1). Fusarium species showed higher inhibition mean value (87%) than *A. flavus* strains (53%). 

##### Propiconazole

The highest propiconazole concentration (250 mg L^−1^) inhibited completely all *F. verticillioides* and *A. flavus* strains. *Fusarium proliferatum* was completely inhibited for a single strain (ITEM 16031), while the two other strains tested had a mean of inhibition of 96% (Appendix A). *Fusarium graminearum* showed mean value of 98% (Figure 1). At the two lowest concentrations of propiconazole, a great variability was observed among fungal species. In particular, *F. verticillioides* was sensitive, with inhibition values of 86% at the lowest concentration (Figure 1), while *F. graminearum*, at the same concentration, was inhibited of 41%. The two species *F. proliferatum* and *A. flavus* showed the same sensitive profile, going through approximately 60, 80, to 100% at the increasing concentrations. 

##### Tebuconazole

Tebuconazole completely inhibited all strains tested at the highest concentration (320 mg L^−1^), up to 10 days of incubation (Appendix A), with the exception of the two *F. graminearum* strains ITEM 126 (inhibition of 93%) and ITEM 6352 (inhibition of 98%) and *F. proliferatum* ITEM 12072 (inhibition of 94%). A greater variability was observed among fungal species at lower concentrations: *F. verticillioides* strains were almost completely inhibited (94%), while *F. proliferatum*, *F. graminearum*, and *A. flavus* showed lower inhibition values: 77%, 67%, and 64%, respectively (Figure 1). 

##### Difenoconazole

Among DMIs, difenoconazole showed the lowest effectiveness. However, after 10 days of incubation, at the highest concentration tested (250 mg L^−1^), *F. verticillioides* and *A. flavus* strains were completely inhibited, while *F. graminearum* and *F. proliferatum* had mean values of inhibition of 77% and 90%, respectively. At the lowest concentration (2.5 mg L^−1^), *F. verticillioides* showed a higher inhibition value (82%) than *F. graminearum* (60%), *F. proliferatum* (67%), and *A. flavus* (52%).

#### 2.1.2. SDHIs

##### Boscalid

Boscalid completely inhibited *A. flavus* mycelial growth at the three concentration tested (500, 50, 5 mg L^−1^), up to 10 days (Figure 1). After three days of incubation (Appendix A), Fusarium species were slightly influenced by the highest concentration (inhibition value up to 30%). However, after 10 days of incubation, all Fusarium species were able to grow in presence of boscalid even at highest concentrations (inhibition values from 0 to 7%).

##### Isopyrazam

Both Fusarium and Aspergillus, except *F. graminearum* (inhibition value of 93%), were completely inhibited by isopyrazam at the highest concentration (200 mg L^−1^) after 10 days of incubation (Figure 1). At the two lowest concentrations (20 and 2 mg L^−1^), a different response was observed among the two fungal genera. *Fusarium graminearum* strains were not affected by the lowest concentration, growing as on unamended control medium, while *F. verticillioides* and *F. proliferatum* showed inhibition values ranging between 2% and 13%, respectively. On media amended with 20 mg L^−1^, all Fusarium species were inhibited with mean values ranging between 24% (*F. graminearum* ITEM 126) and 49% (*F. verticillioides* ITEM 12043). *Aspergillus flavus* was more influenced than the Fusarium species, showing inhibition values of 79% and 89%, at the two lowest concentrations. 

In summary, both SDHI compounds, boscalid and isopyrazam, caused a higher inhibition towards *A. flavus* than towards all Fusarium species tested. 

#### 2.1.3. PPs

##### Fludioxonil

Fludioxonil showed a lower effectiveness compared to other fungicides; after three days of incubation, only *F.graminearum* strains were completely inhibited by the three concentrations tested (50, 5, and 0.5 mg L^−1^), as reported in Appendix A. In *F. verticillioides* species, increasing concentrations of fludioxonil were not positively correlated with inhibition values: *F. verticillioides* ITEM 12052 showed inhibition values of 80%, 70%, and 74% on media amended with 0.5, 5, and 50 mg L^−1^ of fludioxonil, respectively; *F. verticillioides* ITEM 12044 showed the same response of ITEM 12052, with inhibition of 79%, 72%, and 80%. On the other hand, *F. verticillioides* ITEM 12043 was inhibited of about 70% on the three concentrations (Appendix A). After five days of incubation (Appendix A), the colony growth inhibition of *F. graminearum* ranged between 83% and 100%. At the same time, the inhibition of *F. proliferatum* ITEM 12072 was not positively correlated to the increasing concentrations of fludioxonil (Appendix A). Finally, after 10 days of incubation, all Fusarium strains were able to grow on potato dextrose agar (PDA) amended with the three fludioxonil concentrations. Inhibition values, on the highest concentration of fludioxonil, ranged between 54% and 94% for *F. graminearum*, 15% and 28% for *F. proliferatum* and 37% and 50% for *F. verticillioides* (Figure 1). Therefore, a great variability in term of sensitivity to molecule was observed among and within Fusarium species (Figure 1). Similarly, a great variability was observed within *A. flavus* species. At the lowest concentration, *A. flavus* ITEM 8095 showed an inhibition value of 24%, whereas ITEM 8111 and ITEM 8115 showed inhibition values of 54% and 75%, respectively (Appendix A).

#### 2.1.4. MBCs

##### Thiophanate-methyl

Thiophanate-methyl inhibited colony growth of all Fusarium and Aspergillus species tested. Mycelial growth was fully inhibited up to 10 days of incubation, at the highest concentrations (1500 and 150 mg L^−1^). On PDA amended with the lowest concentration (15 mg L^−1^), Fusarium species showed mean values of 93% and 84%, respectively (Figure 1).

#### 2.1.5. Phthalimides

##### Folpet

Folpet was poorly effective to inhibit mycelyal growth of both Fusarium and *A. flavus* species. After three days of incubation (Appendix A), only *A. flavus* and *F. graminearum* ITEM 6352 were completely inhibited at the highest concentration (1200 mg L^−1^). *Fusarium proliferatum* and *F. verticillioides* strains showed inhibition values ranging between 49% (*F. proliferatum* ITEM 12072) and 82% (*F. verticillioides* ITEM 12043). After 10 days, all strains tested were able to grow on PDA amended with the highest dose of the fungicide, with values ranging between 55% (*F. graminearum* ITEM 126) and 78% (*A. flavus* ITEM 8115).

### 2.2. In Vitro Conidial Germination Inhibition

#### 2.2.1. DMIs

*Aspergillus flavus* strains were more inhibited than Fusarium species strains, in conidial germination. *Aspergillus flavus* conidia poorly germinated on media amended with the three different doses of metconazole, prothioconazole, difenoconazole, and prochloraz. On the other hand, Fusarium conidia were able to germinate on media amended with the lowest concentrations of these DMI fungicides, but after 72 h of incubation, the germinated conidia were completely inhibited and germ tube elongation collapsed (Appendix A). 

#### 2.2.2. SDHIs

Boscalid inhibited completely *A. flavus* conidia germination at the three concentrations tested but no effects on Fusarium species were shown (Figure 2). Isopyrazam completely inhibited *A. flavus* conidial germination at the two highest concentrations while at the lowest concentration it inhibited the conidial germination up to 40%. On the other hand, the molecule completely inhibited Fusarium conidial germination at the highest concentration, but it was ineffective at the lowest concentrations (Figure 2).

#### 2.2.3. PPs

Fludioxonil was not effective towards Fusarium species conidial germination, showing a slight inhibition only at the highest concentration (maximum value inhibition of 12% for *F. graminearum* ITEM 6352). On the other hand, fludioxonil showed a good effectiveness against *A. flavus* strains, at the highest doses (mean value of 94% and 68%, respectively), but it was ineffective at the lowest concentration (Figure 2). 

#### 2.2.4. MBCs

Thiophanate-methyl showed values very variable depending on fungal strains (Appendix A), fungal species and fungicide concentration (Figure 2). However, after 72 h, all germinated conidia of *A. flavus* and *Fusarium* strains, except *F. graminearum*, at the lowest concentration (Appendix A), collapsed (Figure 2). Thus, thiophanate-methyl has proved to be active to inhibit both mycelial growth and conidial germination of all strains tested.

#### 2.2.5. Phtalimides

Folpet was the most active among the tested fungicides to inhibit conidial germination (100%) already at the intermediate concentration tested (Figure 2), Folpet showed stronger inhibition towards conidial germination than mycelial growth (Figure 1). At the lowest concentration, only *F. verticillioides* was completely inhibited; *F. graminearum*, *F. proliferatum* and *A. flavus* showed mean values of 7%, 49%, and 90%, respectively.

### 2.3. Fungal Symptoms Assessment on Maize Plants in Field Trials

Infection severity mean values, measured as McKinney index (MKI), evaluated on maize plants are reported in Figure 3.

Fusarium symptoms were observed in all theses, with MKI values ranging between 10.5 (thesis inoculated with *F. proliferatum* and sprayed with prothioconazole) and 49.5 (thesis inoculated with *F. verticillioides* and untreated). Fusarium symptoms were observed even on not inoculated plants, with MKI of 11.1. Highest MKI index were detect on inoculated thesis not sprayed with fungicides (MKI of 45.1, 48.5, and 49.5 for *F. graminearum*, *F. proliferatum*, and *F. verticillioides*, respectively). The plants treated with prothioconazole showed the lowest MKI with a value of 13.2, 10.5, and 19.7 for *F. graminearum*, *F. proliferatum* and *F. verticillioides*, respectively.

Symptoms of Aspergillus infections were not observed on the maize ears.

### 2.4. Re-Isolation of Fungal Species from Maize Plants by Mycological Analyses

The fungal colonies, originated from 100 representative kernels for each field trial sample, were identified after five days of incubation, based on morphological characteristics. From samples inoculated by a given *Fusarium* species, the strains re-isolated belonged to the *Fusarium* species used for the inoculation; from the not inoculated and untreated samples, *F. verticillioides* and *F. proliferatum* were isolated (20% and 2%, respectively).

Fusarium contamination was detected in all the maize samples, ranging from 22% (not inoculated thesis) to 95% (untreated thesis inoculated with *F. verticillioides*), as shown in Figure 3. Among Fusarium species, a different capability to colonize kernels has been observed. The lowest fungal contamination was detected in the species inoculated with *F. graminearum* (34–70%). *Fusarium verticillioides* was detected in all theses with high contamination values, ranging between 82% and 95%. *Fusarium proliferatum* was detected with values ranging between 39% and 70%.

Contamination of Aspergillus species, mainly belonging to section Nigri (90%), was detected in all species (data not shown). The values were 24% in not inoculated and untreated species, 15% in theses inoculated but untreated and in species treated with thiophanate-methyl, and 40% in species treated with prothioconazole. In particular, when *F. graminearum* was inoculated, a higher presence of Aspergillus species was detected (12%), while when *F. verticillioides* was inoculated, Aspergillus contamination was negligible (0.5%). Values of 8% were detected inoculating *F. proliferatum*.

In the experiment addressed to *Aspergillus flavus* (Figure 4), strains belonging to this species were detected in all inoculated theses, with values ranging between 7 (theses treated with boscalid), a value very similar to the not inoculated thesis (4.9%), and 25.7 (untreated thesis). However, in all the samples, also Fusarium contamination was detected, with values ranging between 2% (theses treated with prothioconazole) and 15.5% (untreated thesis not inoculated with *A. flavus*). 

## 3. Discussion

This study provides new information on the sensitivity of the main toxigenic Fusarium species associated to FER of maize and *A. flavus*, to different fungicides. Fusarium maize ear rot is a major problem in temperate areas worldwide. In the last years, several guidelines have been suggested to optimize cropping systems in order to minimize fungal and consequent mycotoxin contamination [20]. However, when pedoclimatic conditions favorable to fungal disease development occur, direct control of fungi through use of fungicides could be an option to consider. On the other hand, chemical protection to control FER is not widely used yet and few studies have demonstrated the efficacy of synthetic fungicides on reduction of the fungal species associated to FER in maize [23,33]. Thus, studies on the effectiveness of novel fungicides, or fungicides registered on other crops, for controlling FER are highly useful. The use of different fungicides, mainly belonging to DMI compounds, is allowed to control disease associated to Fusarium species on wheat and other minor cereals [27,34]. In the present study, we tested the in vitro efficacy of eleven fungicides, belonging to DMI, SDHI, PP, MBC, and Pthalimides fungicides, among which folpet displays a multisite activity. Target-site fungicides provide several advantages when compared to multisite ones, such as a high persistence in plant tissues, that allows to reduce the number of treatments for crop season; higher effectiveness at lower doses; and higher selectivity against fungal pathogens [35,36].

We demonstrated that all tested DMIs reduced fungal development with higher efficacy than the other chemicals tested. However, a certain variability in their activity was observed. In particular, prothioconazole and prochloraz proved to be the most active molecules against both Fusarium species and *A. flavus*.

Previous studies showed that prochloraz was effective against *F. culmorum* and *F. langsethiae* [37,38]. Tebuconazole and prothioconazole showed effectiveness against a wide range of Fusarium species associated with FHB (*F. avenaceum*, *F. culmorum*, *F. graminearum*, *F. poae*, *F. tricinctum*, *F. sporotricioides*, and *F. crookwellense*) [39,40]. Moreover, the effectiveness of DMI fungicides has been demonstrated also for *A. flavus* strains, as reported by Formenti et al. [41], that demonstrated the sensitivity of *A. flavus* to prochloraz. On the other hand, the repeated use of chemicals with the same mode of action could lead to a strong selection of resistant strains. Indeed, decreased sensitivity to tebuconazole has been detected in Germany [42] and China, where DMI fungicides have been largely used in the last 30 years [32]. Since azoles are considered the most prominent fungicide class used in cereals up to date, the possible development and accumulation of resistance in fungal populations increases the need of identifying new chemicals provided of different modes of action not yet registered on maize, such as SDHI, MBC, and PP fungicides.

The recently introduced SDHI fungicides, such as boscalid and isopyrazam, have shown a great efficacy against a large spectrum of fungi [43]. However, a few years after the introduction of SDHIs, resistance has been reported in field populations and laboratory-induced mutant strains of several phytopathogenic fungi [43]. Neverthless, a not generalized cross-resistance has been observed: for instance, mepronil-resistant *Rhizoctonia solani* strains showed sensitivity to boscalid [44], *Corynespora cassiicola* and *Podosphaera xanthii* strains were reported to be highly resistant to boscalid but were sensitive to fluopyram [45], and field strains of *Botryotinia fuckeliana* were resistant to boscalid and sensitive to fluopyram [46]. In several phytopathogenic fungal species, the resistance to SDHIs has been identified in their ability to substitute some amino acids in the target proteins. 

In order to better elucidate this mechanism of resistance, preliminary investigations on the molecular characterization of the four genes encoding succinate dehydrogenase subunits indicated that a high variability exists between Aspergillus and Fusarium genera, which is likely related to their different response (Masiello, data not shown).

The efficacy of some molecules belonging to SDHIs against other Fusarium species has been demonstrated in China, where *F. asiaticum* (a member of *F. graminearum* species complex [47]) strains were sensitive to the novel SDHI pydiflumetofen [48]. Moreover, in Southern Brazil, SDHIs in mixture with quinone outside inhibitors fungicides (QoI) were able to reduce *F. graminearum* on wheat in field conditions [49].

On the contrary, other studies demonstrated the insensitivity of *F. graminearum* to the SDHI isopyrazam as well as to QoI trifloxystrobin, suggesting that fungal respiration in *F. graminearum* seems to be significantly different from other Fusarium [50,51].

In our studies a high resistance of all Fusarium strains to boscalid and sensitivity to the highest concentration of isopyrazam were detected. 

The MBC thiophanate-methyl inhibited mycelial growth and conidial germination of *Fusarium* species and *A. flavus*. In the last decades, the MBC carbendazim has been widely used to control Fusarium diseases on several plants, including wheat, tomato and cyclamen, showing a great capability to prevent Fusarium infections. However, several studies reported the occurrence of resistant field strains to MBCs [30,35,52]. Therefore, since thiophanate-methyl is not authorized on maize crop, and resistance issues do exist, further studies aimed to confirm its possible use in the control of toxigenic fungi on maize should be carried out. 

Among PPs, fludioxonil, used alone or in mixture with other fungicides, has been largely used for cereals and soybean seed treatment, showing a high efficacy against Fusarium species [53,54,55,56]. We showed that all Fusarium and Aspergillus strains tested were slightly inhibited by this compound, and fungal growth inhibition was not correlate to the increasing fludioxonil concentrations. Moreover, some Fusarium strains (*F. graminearum* ITEM 6415, *F. proliferatum* ITEM 12052 and ITEM 12072, and *F. verticillioides* ITEM 12043 and ITEM 12044) acquired the ability to grow on fludioxonil amended media, showing sectorization of mycelium with new phenotypic traits, retained when the strains were re-inoculated on fresh PDA amended with fludioxonil, as previously reported by Broders et al. [54]. A great variability of response to fludioxonil exists among and within *Fusarium* species. Peters et al. [57] reported that reference strains of *F. sambucinum* and *F. coeruleum* were sensitive to fludioxonil, but all tested field strains of the same two species were resistant to fludioxonil, showing no growth inhibition up to 100 mg L^−1^ of molecule concentration. On the other hand, the efficacy of fludioxonil has been demonstrated by Ellis et al. [58], since soybean seeds treated with phthalimides captan and fludioxonil had the lowest disease severity and *F. graminearum* symptoms, compared to azoxystrobin use. Also, the efficacy of fludioxonil in mixture with the Phenylamides metalaxyl-M against *F. verticillioides* has been demonstrated by Miguel et al. [59]. 

The in vitro tests on molecules available on the market allowed us to select the most effective molecules against the most occurring mycotoxigenic fungi on maize. In particular, prothioconazole and thiophanate-methyl effectively inhibit *Fusarium* species, while prothioconazole and boscalid completely inhibit *A. flavus* (Figure 1 and Figure 2). 

Since in field conditions several variables interfere with fungal infection or disease development, subsequent experiments in field have been carried out to confirm in vitro tests. Prothioconazole confirmed the efficacy to control *F. graminearum* and *F. proliferatum*, as shown in Figure 3, showing a reduction on fungal contamination of 51% for *F. graminearum* and 44% for *F. proliferatum*, compared to untreated thesis. On the contrary, *F. verticillioides* contamination was similar in all the theses, suggesting a great capability of this species to colonize maize plants despite environmental conditions or chemical treatments. Moreover, if we consider the severity of fungal infection, it is very interesting to notice that MKI values of theses treated with prothioconazole are about 14%, compared to 48% of inoculated untreated theses (Figure 3). 

Likewise, the application of thiophanate-methyl caused a reduction of both *F. graminearum* and *F. proliferatum* of 49% and 28%, respectively, while it is noneffective to control *F. verticillioides*. 

Prothioconazole application on maize plants was effective also to control *A. flavus*, although less than Fusarium species, causing a reduction of 51%. This species is almost completely controlled by boscalid, that reduced the contamination in kernels of 73% (Figure 4).

As expected, the performance of fungicides in planta showed a lower efficacy if compared to tests in vitro, although the results obtained with the two different approaches were similar. However, Fusarium contamination was detected even in plants not inoculated with Fusarium species or with *A. flavus* (Figure 3 and Figure 4) and Aspergillus was detected also in the Fusarium experimental trial, probably due to cross-contamination or natural seeds contamination. Indeed, the presence of Aspergillus species, mainly belonging to section Nigri, when Fusarium infection is reduced (theses not inoculated or treated with fungicides) and vice versa, can be explained by the different ability of fungal species to compete in infecting host plants. 

Probably, the tested dose of single fungicides was sublethal for fungi, since it can be suggested to increase doses to better control mycotoxigenic fungi on maize. Moreover, a further treatment, in addition to the application at flowering, could help to reduce fungal contamination during crop season. However, the mixture of fungicides with different modes of action is the best option to control the whole population of mycotoxigenic fungi occurring on maize, limiting, at the same time, the development of resistance in field strains.

## 4. Conclusions

This study has allowed to obtain useful information about the sensitivity of *F. graminearum*, *F. proliferatum*, *F. verticillioides*, and *A. flavus* species, strictly associated to maize diseases and mycotoxicological risk, to the main fungicides nowadays registered on the most important crops. 

In vitro experiments showed that all SDHIs had a very high efficacy against *A. flavus*, while only the highest concentration tested of isopyrazam was effective against Fusarium species. For this reason, this class of fungicides could improve the control of the main toxigenic fungi contaminating maize only if applied in mixture with molecules belonging to DMI or MBC. All DMIs, in particular prothioconazole and prochloraz, and the MBC thiophanate-methyl showed the best effectiveness to inhibit both Fusarium species and *A. flavus* in vitro. 

The activity of the prothioconazole and thiophanate-methyl has been evaluated both in vitro and in field conditions, where several pedoclimatic and agricultural parameters could influence the host–pathogen interaction. Indeed, a different performance among Fusarium species was observed in field conditions, since *F. verticillioides* was not inhibited.

These compounds are not authorized on maize crop, and studies on their effectiveness in controlling FER and *A. flavus* contamination in maize field conditions are limited. Thus, this study could be useful to select the best molecules active against Fusarium and Aspergillus species associated with maize diseases, for further investigations on their feasibility.

## 5. Materials and Methods

### 5.1. Fungal Strains

The effectiveness of chemical compounds was evaluated against *F. graminearum, F. proliferatum*, *F. verticillioides*, and *A. flavus* species; for each fungal species three strains were tested. All strains were obtained from the ITEM Fungal Collection of the Institute of Sciences of Food Production, Bari (www.ispa.cnr.it/Collection). In detail, *F. graminearum* strains (ITEM 126, ITEM 6352, and ITEM 6415) were isolated from durum wheat in Northern Italy; *F. proliferatum* (ITEM 12072, ITEM 12103, and ITEM 16031), *F. verticillioides* (ITEM 12043, ITEM 12044, and ITEM 12052), and *A. flavus* strains (ITEM 8095, ITEM 8111, and ITEM 8115) were all isolated from maize kernels in Northern Italy. 

All fungal strains were refreshed on Petri dishes (90 mm in diameter) containing potato dextrose agar (PDA Oxoid CM0139), incubated at 25 ± 1 °C under an alternating light/darkness cycle of 12 h photoperiod.

### 5.2. Fungicides Tested In Vitro

Among the most effective molecules currently used to control fungal diseases caused by ascomycetes fungi, eleven fungicides, belonging to 8 different chemical classes and showing 5 different modes of action, were tested (Table 1).

In particular, 2 fungicides—boscalid (Cantus) and isopyrazam (Zulu)—belong to SDHIs and are not registered on cereals. Six fungicides, registered on cereals, belong to DMIs: metconazole (Caramba), difenoconazole (Score25 EC), propiconazole (Opinion Ecna), and tebuconazole (Icarus EW), grouped as triazoles, the imidazoles prochloraz (Carnival) and the triazolinthiones prothioconazole (Proline). Fludioxonil (Celest) belongs to PPs, and is registered in cereals for seed coating. Thiophanate-methyl (Enovit Metil) belongs to MBCs, and folpet (Folpan80) belongs to Phtalimides fungicides; both were never registered on cereals.

The authors have mentioned the trade names of the tested fungicides for the scientific purpouse and this does not reflect any recommendation for use.

Based on dose recommended in label by manufacturers, we tested, for each fungicide, the manufacturers’ suggested concentration and the two lower decimal dilutions, as reported in Table 1. 

Fungicides were suspended in sterile distilled water and added to sterilized PDA or Water Agar (WA, 20 g L^−1^ agar Oxoid n. 3) media, cooled down to 45–50 °C. 

### 5.3. Mycelial and Conidial Germination Assays

The activity of fungicides against the fungal strains was evaluated by measuring colony growth and conidial germination.

Each fungicide was suspended in distilled sterile water to obtain three different concentrations to be tested. The highest concentration was prepared in order to represent the effective dose of the active ingredient, recommended by the manufacturer for field treatments; the subsequent two concentrations were in a ratio 1:10 to each other. Appropriate volumes of liquid fungicides were added to sterile PDA cooled down to 45–50 °C, in order to obtain the concentrations shown in Table 1. Each active ingredient was tested in three concentrations against fungal species strains, in triplicate. In the control theses only PDA was used.

In mycelial growth assay, mycelium disks (4 mm in diameter) from actively growing margins of 3–5 day old colonies cultured on PDA, were used to inoculate Petri dishes (90 mm in diameter) containing PDA or PDA amended with fungicides. The inhibition activity of the fungicide on colony growth was determined by measuring the diameter (in mm) of developing colonies after 3, 5, 7, and 10 days of incubation at 25 ± 1 °C, under an alternating light/darkness cycle of 12 h photoperiod. A ruler was used to measure the two orthogonal diameters of the colony. The inhibition caused by each fungicide concentration was expressed as percentage value, reporting the difference between the maximum level of inhibition with no growth of the fungal colony on the medium (100%), and the ratio between diameter of colony growth on PDA amended with the fungicide and the diameter of the growth of colony inoculated on PDA medium as control (value of 0% means the total lack of inhibition corresponding to full growth of the fungal colony on the medium). The mean values obtained from the three tested strains were reported for each species.

In conidial germination test, conidia were collected by scraping the surface of 7-day-old colonies, grown on PDA at 25 ± 1 °C under an alternating light/darkness cycle of 12 h photoperiod, with a sterilized loop, suspended in sterile distilled water containing 0.05% Tween 20, and filtered through Miracloth (Calbiochem, La Jolla, Canada) to remove mycelia fragments. Aliquots (10 μL) of conidial suspension (1 × 10^5^ conidia mL^−1^) were spotted on disks (6 mm diam) of WA medium and WA amended with fungicides (3 concentrations, as reported above for mycelium growth experiments) placed on sterile microscope slides. The disks were incubated in a moist chamber at 25 ± 1 °C in darkness and after 48 h; 100 conidia randomly selected on each of three replicated spots were observed at optical microscope with 125× magnification, germinated conidia were counted. After 72 h, the ability of conidia to growth was confirmed observing the germ tube elongation. The frequency of conidia able to germinate on medium amended with fungicide was calculated considering the frequency of conidia germination on the untreated control medium. The mean values obtained from the three tested strains were reported for each species.

### 5.4. Field Experiments Settings

Based on in vitro test results, the most effective fungicides were selected to set experiments on maize plants in field. Prothioconazole, as representative of DMIs, and the MBC thiophanate-methyl were tested against *F. graminearum*, *F. proliferatum*, and *F. verticillioides*; prothioconazole and boscalid (SDHI) against *A. flavus*.

Uncoated seeds of the commercial maize variety “Marano 0501” were selected for the experiment. A representative seed sub-sample was previously analyzed to confirm the low Fusarium and Aspergillus species contamination, in order to exclude possible source of fungal contamination during experimental trial.

Two experimental trials were separately set up to test the effectiveness of fungicide application during flowering stage against Fusarium species and *A. flavus*.

For each fungal species 4 different theses were compared, including a negative control (plants not inoculated), a positive control (plants inoculated and untreated), and two theses treated with the two selected fungicides. Each thesis consisted of 24 maize plants, in particular, 3 replicates of 8 plants were considered, following a randomized block experimental design. Each plot, of 125 × 210 cm, was separated by a border line plants in order to exclude cross contamination. The experimental designs used for Fusarium and Aspergillus trials are detailed in Appendix A.

#### 5.4.1. Fungicide Application and Fungal Inoculum

Two days before fungal inoculation, each plot, each well separated with plastic films in order to exclude cross contamination, was sprayed with fungicide arranging the dose recommended by manufacturers for other crops: 0.8 L/Ha for prothioconazole (Proline), 1.25 L/Ha for thiophanate-methyl (Enovit Methyl), and 1 Kg/Ha for boscalid (Cantus).

In pre-flowering time, conidial suspensions in sterile water containing 10^−5^ conidia mL^−1^ of each Fusarium and Aspergillus species were prepared by scraping the surface of 7-day-old colonies grown on PDA, and then stored at 4 °C until use. At flowering time and two day after fungicide application, 500 µL of each conidial suspension were sprayed, according to the planned theses, on flowery silks of each maize ear, by using an atomizer.

#### 5.4.2. Evaluation of Fungal Symptoms on Maize Plants

At harvesting time, symptoms of fungal infections on maize plants were assessed according to an empirical scale with seven classes of severity, from 0 (absence of infections) to 6 (extensive symptoms of infection closed to 100% of plant surface). Infection severity was calculated by McKinney Index (MKI).

#### 5.4.3. Fungal Re-isolation and Growth Conditions

At harvesting time, each plot consisting of 8 maize plants was harvested and stored at 4 °C. For each sample, 100 representative kernels were superficially disinfected in a 2% sodium hypochlorite solution for 2 min, washed twice with sterile distilled water for one minute and then plated (10 kernels/plate) on PDA added with pentachloronitrobenzene (PCNB; 500 mg L^−1^), streptomycin (100 mg L^−1^), and neomycin (50 mg L^−1^). After 5 days of incubation at 25° ± 1 °C under fluorescent light (12 h photoperiod), the contamination of the inoculated fungal species was detected for each sample. In addition, in the samples inoculated with *A. flavus*, both *A. flavus* and Fusarium contamination was evaluated, in order to evaluate the effectiveness of fungicides against the natural Fusarium contamination occurring on maize seeds.

## Figures and Tables

**Figure 1 toxins-11-00011-f001:**
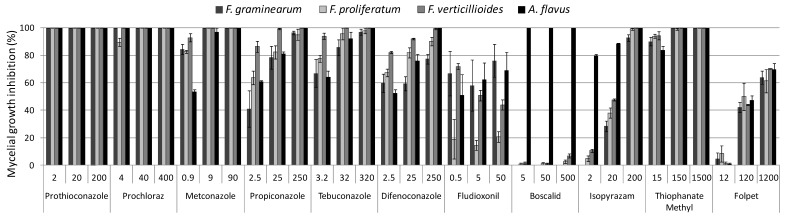
Colony growth inhibition on potato dextrose agar (PDA), amended with three different concentrations of fungicides (expressed in mg L^−1^ and reported on the x-axis), after 10 days of incubation at 25 °C. Percentage values calculated on the colony growth (expressed in mm) of the control thesis for each species, as reported in Material and Methods.

**Figure 2 toxins-11-00011-f002:**
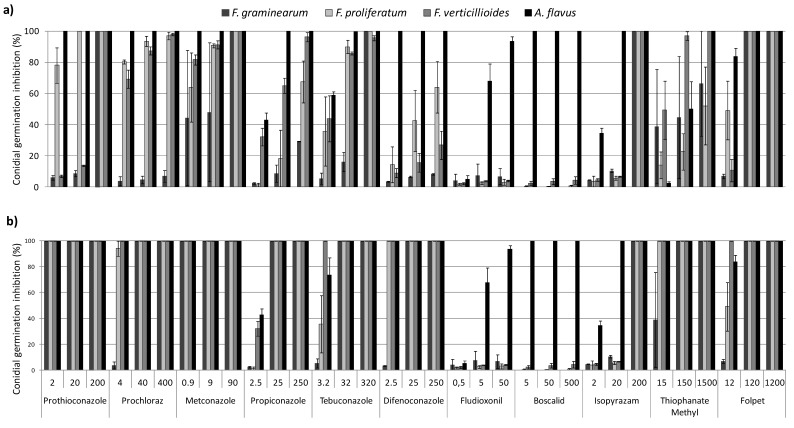
Conidial germination inhibition on water agar amended with three different concentrations of fungicides (expressed in mg L^−1^ and reported on the x-axis) (**a**) after 48 h of incubation at 25 °C and (**b**) after 72 h of incubation at 25 °C. Percentage values calculated on the conidial germination of the control thesis for each species, as reported in Material and Methods.

**Figure 3 toxins-11-00011-f003:**
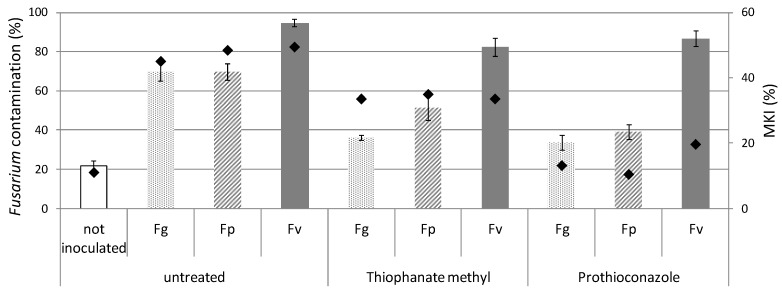
Fusarium contamination evaluated on maize samples not inoculated and inoculated with *Fusarium graminearum* (Fg), *F. proliferatum* (Fp), and *F. verticillioides* (Fv) and untreated or treated with thiophanate-methyl and prothioconazole. Error bars represent standard error among replicates. The black points show the Mc Kinney Index (MKI).

**Figure 4 toxins-11-00011-f004:**
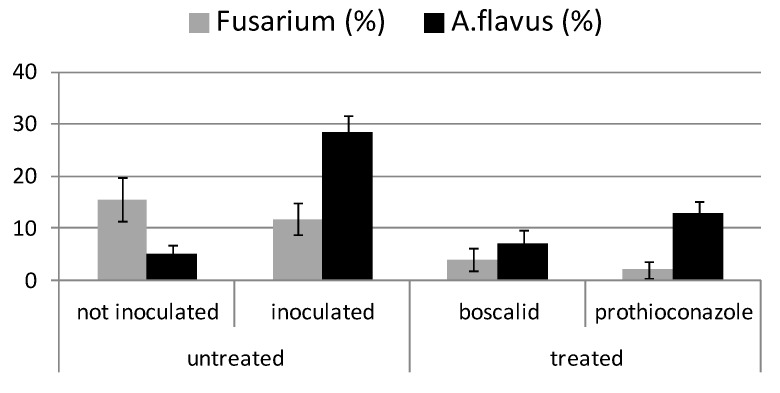
*Aspergillus flavus* contamination evaluated on maize samples inoculated and not inoculated with *A. flavus* in field trial. Fusarium contamination was also detected as natural contamination. Error bars represent standard error among replicates.

**Table 1 toxins-11-00011-t001:** Fungicide tested in colony growth and conidial germination assays.

Commercial Name	Active Ingredient	Active Ingredient Tested (mg L^−1^)	Chemical Group *	Group Name *	Target Site *	Mode of Action *
Cantus	Boscalid	500–50–5	pyridine-carboxamides	SDHI (Succinate dehydrogenase inhibitors)	complex II: succinate-dehydrogenase	Respiration
Zulu	Isopyrazam	200–20–2	pyrazole-4-carboxamides			
Carnival	Prochloraz	400–40–4	Imidazoles	Demethylation Inhibitors SBI Class I	C14-demethylase in sterol biosynthesis	Sterol biosynthesis in membranes
Proline	Prothioconazole	200–20–2	Triazolinthiones			
Icarus	Tebuconazole	320–32–3.2	Triazoles			
Opinion Ecna	Propiconazole	250–25–2.5				
Caramba	Metconazole	90–9–0.9				
Score	Difenoconazole	250–25–2.5				
Celest	Fludioxonil	50–5–0.5	phenylpyrroles	PP-fungicides (PhenylPyrroles)	MAP/Histidine—Kinase in osmotic signal transduction	signal transduction
Enovit Metil FL	Thiophanate-methyl	1500–150–15	Thiophanates	MBC-Fungicides (Methyl Benzimidazole Carbamates)	β-tubuline assembly in mitosis	mitosis and cell division
Folpan80	Folpet	1200–120–12	phthalimides	Phthalimides	multisite contact activity	Multisite contact activity

* Information from Fungicide Resistance Action Committee (FRAC), available at www.FRAC.info.

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
