# Peer review of "In Vitro and in Field Response of Different Fungicides against Aspergillus flavus and Fusarium Species Causing Ear Rot Disease of Maize"

_toxins, 2019, doi:10.3390/toxins11010011_

Round 1
Reviewer 1 Report
The manuscript entitled “In vitro and in field response of different fungicides active against Aspergillus flavus and Fusarium species causing ear rot disease of maize” describes the effect of fungicides towards various pathogenic fungi. Interestingly, several types of fungicides were tested and several of them show interesting results, in vitro, depending on tissues, and in vivo. From the materials and methods it seems the experimental setup was correct. Overall, the reading and interpretation of the manuscript is the difficult part since the language is not always correct or well interpretable. The authors seem to have a problem with comparisons. Several of them are mentioned below, as well as other suggestions for corrections/modifications. Due to the shear amount of English improvements needed it is recommended to incorporate a check of a native writer. Similarly, the discussion would benefit from rewriting and avoiding the currently abundantly used woolly phrases. It is quite long and is quite a repeat of the results. Please focus on comparisons with other literature and maybe on the explanation for the differences found between previous literature or efficacy differences between tissues. Similarly, large improvements can be made for the M&M section.
Remarks/examples
Line 2-Suggestion to remove “active” from the title. It does not contribute to the title and is questioning what is an effective or ineffective fungicide.
Line 8-please rewrite “being few molecules authorised”.
Line 15-you cannot phrase that the compounds “confirmed their efficacy”
Line 22-“different” is quite meaningless, please avoid and rewrite
Line 35- use “can accumulate”
Line 52-suggestion to write “higher temperatures” or “elevated temperatures”
Then in the result section, there are many comments that all are based on a kind or comparison e.g:
Line 115-rewrite “was very moderately less”??
Line 128-rewrite “Resulted the less effective”??
Line 132- rewrite “resulted more sensitive”
Line 138- “resulted not able”??
Line 140- “Resulted able to”
Line 150- “were more effective”, in this case one wonders compared to what it is more effective....
Other issues
Line 154, Table S1 (introduce a space)
Line 156/7, please complete the L-1 on the same line
Line 173- “The value of mycelial growth inhibition” please rewrite like “Mycelial growth was fully inhibited up to ..”
Line 185 & 221/222, In the figures 1 & 2, there is no control present or indicated and nowhere it is explained what the mycelial growth inhibition is based on.
Figure 1 and 2, what are the numbers meaning for the fungicides, mM, ug? nM?.mg/L?
Line 199, (note again, these are just examples, screen the whole manuscript for similar issues) “the product resulted able” ????
Line 215, based on what is Folpet most active, since it also has the highest concentrations applied (Figure 2, up to 1200) how can you say this? It is the only component in this figure with such high concentration. If you relate to mycelial growth than write it properly.
Line 235, what does “after 5 days incubation” mean. 5 days after inoculation? These are field trials, so what is incubation meaning?
Line 251- What is “Intermediate values were detected inoculating F. proliferatum” pointing at?
Line 255, Since the Mc Kinney Index is unrelated from one sample to the other, it is not justified to make it as line connected dots.
Line 267 “new information” (not plural)
Line 332- Typo “Fusariun”
Please reduce the number of “In our study”, “The present study”, etc..
Line 428- Rewrite “According to recently guidelines”
Another example, Line 466 “The response to fungicides of the fungal strains tested”. This phrase suggests that fungal strains are producing fungicides...??
Line 473, “The three concentrations of active ingredient for each fungicide” might suggest that three active ingredients are merged in one fungicide but this is not the case..
Author Response
Comments and Suggestions for Authors
The manuscript entitled “In vitro and in field response of different fungicides active against Aspergillus flavus and Fusarium species causing ear rot disease of maize” describes the effect of fungicides towards various pathogenic fungi. Interestingly, several types of fungicides were tested and several of them show interesting results, in vitro, depending on tissues, and in vivo. From the materials and methods it seems the experimental setup was correct. Overall, the reading and interpretation of the manuscript is the difficult part since the language is not always correct or well interpretable. The authors seem to have a problem with comparisons. Several of them are mentioned below, as well as other suggestions for corrections/modifications. Due to the shear amount of English improvements needed it is recommended to incorporate a check of a native writer. Similarly, the discussion would benefit from rewriting and avoiding the currently abundantly used woolly phrases. It is quite long and is quite a repeat of the results. Please focus on comparisons with other literature and maybe on the explanation for the differences found between previous literature or efficacy differences between tissues. Similarly, large improvements can be made for the M&M section.
First of all, we would like to thank the referee for the precious suggestions and criticisms highly useful to improve our paper.
We revised all the manuscript to improve language and phrasing and to facilitate reading. We have made all the indicated corrections/modifications, according with reviewer indications.
Remarks/examples
Line 2-Suggestion to remove “active” from the title. It does not contribute to the title and is questioning what is an effective or ineffective fungicide.
We removed the word.
Line 8-please rewrite “being few molecules authorised”.
Replaced with “since few molecules are registered”
Line 15-you cannot phrase that the compounds “confirmed their efficacy”
We have rewritten the sentence.
Line 22-“different” is quite meaningless, please avoid and rewrite
Line 23 new version - We have modified.
Line 35- use “can accumulate”
Line 37 - Done
Line 52-suggestion to write “higher temperatures” or “elevated temperatures”
Line 54 - We have written “higher temperatures”.
Then in the result section, there are many comments that all are based on a kind or comparison e.g:
We revised the phrases for comparisons in the manuscript.
Line 115-rewrite “was very moderately less”??
Line 121 - Rewritten.
Line 128-rewrite “Resulted the less effective”??
Lines 135-136 - Rewritten.
Line 132- rewrite “resulted more sensitive”
Lines 139-140 - Rewritten.
Line 138- “resulted not able”??
Line 147 - We have modified.
Line 140- “Resulted able to”
Line 150 - We have modified.
Line 150- “were more effective”, in this case one wonders compared to what it is more effective....
Line 161 - We changed the sentence; however the comparison was between Aspergillus and Fusarium and not with other fungicides.
Other issues
Line 154, Table S1 (introduce a space)
Line 167 - Done.
Line 156/7, please complete the L-1 on the same line
Line 170 - Done.
Line 173- “The value of mycelial growth inhibition” please rewrite like “Mycelial growth was fully inhibited up to ..”
Line 187-188 - Done
Line 185 & 221/222, In the figures 1 & 2, there is no control present or indicated and nowhere it is explained what the mycelial growth inhibition is based on.
Line 200 & 237 - Sorry, if figures were not clear enough. However, we have clearly reported in M&M that the values were expressed in % of colony growth with respect to the control colony growth, as reported in the following lines: 509-514. Nevertheless, we added a sentence in the figure caption.
Figure 1 and 2, what are the numbers meaning for the fungicides, mM, ug? nM?.mg/L?
Line 200 & 237 - We reported mg L-1 in the figure captions.
Line 199, (note again, these are just examples, screen the whole manuscript for similar issues) “the product resulted able” ????
Line 216 - We changed similar phrases in all the manuscript
Line 215, based on what is Folpet most active, since it also has the highest concentrations applied (Figure 2, up to 1200) how can you say this? It is the only component in this figure with such high concentration. If you relate to mycelial growth than write it properly.
Line 232 - We tested the manufacturers’ suggested concentration for all the fungicides. Then we tested also two lower concentrations diluted 1:10 for all fungicides. Thus, Folpet has such high concentration, but not the highest, because it is the suggested concentration by the manufacturers.
Line 235, what does “after 5 days incubation” mean. 5 days after inoculation? These are field trials, so what is incubation meaning?
Line 254 - This is the material collected from the field, from which the fungi were re-isolated. As we explained in M&M, paragraph 5.4.3, we collected kernels from all the plants and we analyzed representative samples to re-isolate fungal species from maize plants. Thus, these kernels were plated on a medium to allow the growth of fungal colonies after incubation.
Line 251- What is “Intermediate values were detected inoculating F. proliferatum” pointing at?
Line 270 - We reported the precise value also for F. proliferatum.
Line 255, Since the Mc Kinney Index is unrelated from one sample to the other, it is not justified to make it as line connected dots.
Line 272 - We changed the kind of graph for the MKI.
Line 267 “new information” (not plural)
Line 291 - We corrected
Line 332- Typo “Fusariun”
Corrected
Please reduce the number of “In our study”, “The present study”, etc..
We reduced these phrases in the manuscript.
Line 428- Rewrite “According to recently guidelines”
We eliminated this part from conclusions.
Another example, Line 466 “The response to fungicides of the fungal strains tested”. This phrase suggests that fungal strains are producing fungicides...??
Line 456 - Sorry, we didn’t understand such remark. However we changed that sentence.
Line 473, “The three concentrations of active ingredient for each fungicide” might suggest that three active ingredients are merged in one fungicide but this is not the case..
Line 463 - We changed the sentence.

Reviewer 2 Report
The submitted work provides important information on the effect of different fungicides on several plant pathogenic and toxigenic fungi. The in vitro trials were followed by field trials to confirm their efficacy in the field. The stated outcomes of this research will be used as useful information for the registration of chemical compounds on maize. One of the main important parameters that should have been investigated is the effect of the tested fungicides on mycotoxin production. As stated in the literature, production of mycotoxins is sometimes enhanced, especially in case of Fusarium species, under stress conditions. Also the same happens with application of fungicides and biocontrol agents to control the fungus. So, while the fungal growth is inhibited, mycotoxin production “in many cases” is enhanced. Analysis of the main mycotoxins and their masked forms (such as DON3G, ZEN14S etc in case of DON and ZEN) should have been considered in the field trials.
General comments
Although the manuscript is good written especially the introduction part, the language should be checked by a professional or native speaker.
For the commercial name of the fungicides, does not make a conflict of interest for mentioning the trade names? I would suggest to either remove the trade names OR to mention that “The authors have mentioned the trade names of the tested fungicides for the scientific purpose and this does not reflect any recommendation for use”.
The Introduction part
Line 67 and 68….. the authors referred to some papers used new hybrids and swing time, it would be better to refer also to papers focusing on biological control of toxigenic fungi and their toxins in the pre-harvest stage. As an example “Biological control of mycotoxigenic fungi and their toxins: An update for the pre-harvest”. This is a new published work could be helpful.
Results
According to Figure 1, Prochloraz was also among the best fungicides controlling the fungal growth. Is there a specific reason why the authors did not include this fungicide in the field trial as well?
Line 115, what do you mean by “very moderately” ? please replaced it with an appropriate adjective.
Line 135, please replace the word “blocked” with an appropriate verb
Line 138, please rewrite again
Line 140 , please rewrite again
Line 148, not correct English,,, please rewrite
162 what do you mean by “positively inhibited”. Please replace it with another word.
Line 172 replace the word “block”
Figure 4, why did you combined all the fusarium together ? It will be better if the authors can split the Fusarium species rather than combining all the species.
Discussion
Since the authors didn’t investigate the mycotoxins, therefore please focus only on the fungus not the toxins because the production of the mycotoxins (either suppression or enhancement is unknown).
The discussion is too long and needs to be shortened to avoid the confusion may happen to the readers.
Materials and methods
Line 446, “probably never or rarely exposed”, what do you mean ? either it is exposed or not ! please make sure about the information and restate it.
Line 447, the concentration of the PDA is the same concentration stated by the manufacturer. Please delete it as long as there is not difference or replaced it by “according to manufacturer’s instruction”
Line 448, how did you grow the fungi ? in Petri dishes ? which size.. 90 mm ? please state.
Table 1, Please provide the references “if any” for the information you state ?
Line 463, did you autoclave the fungicides ? what about their stability at such a high temperature? If not then please write the sentence again.
Line 478 how did you measure the diameter ? by a ruler or a digital calibre ? please state either in cm or mm
Line 494, what kind of microscope did you used ? please state
Line 523, what media did you use to store the conidia ? sterile water ? or PBS? Please state
Author Response
Comments and Suggestions for Authors
The submitted work provides important information on the effect of different fungicides on several plant pathogenic and toxigenic fungi. The in vitro trials were followed by field trials to confirm their efficacy in the field. The stated outcomes of this research will be used as useful information for the registration of chemical compounds on maize. One of the main important parameters that should have been investigated is the effect of the tested fungicides on mycotoxin production. As stated in the literature, production of mycotoxins is sometimes enhanced, especially in case of Fusarium species, under stress conditions. Also the same happens with application of fungicides and biocontrol agents to control the fungus. So, while the fungal growth is inhibited, mycotoxin production “in many cases” is enhanced. Analysis of the main mycotoxins and their masked forms (such as DON3G, ZEN14S etc in case of DON and ZEN) should have been considered in the field trials.
First of all, we would like to thank the referee for the precious suggestions and criticisms highly useful to improve our paper.
This paper is focused on fungal response against fungicides in order to reduce the presence of the toxigenic fungi in the field though the chemical control. We have limited the comments on mycotoxins.
General comments
Although the manuscript is good written especially the introduction part, the language should be checked by a professional or native speaker.
For the commercial name of the fungicides, does not make a conflict of interest for mentioning the trade names? I would suggest to either remove the trade names OR to mention that “The authors have mentioned the trade names of the tested fungicides for the scientific purpose and this does not reflect any recommendation for use”.
We left the commercial names because we tested the manufacturers’ recommend doses against the fungal strains. Thus, we mentioned the sentence suggested by the referee in the manuscript.
The Introduction part
Line 67 and 68….. the authors referred to some papers used new hybrids and swing time, it would be better to refer also to papers focusing on biological control of toxigenic fungi and their toxins in the pre-harvest stage. As an example “Biological control of mycotoxigenic fungi and their toxins: An update for the pre-harvest”. This is a new published work could be helpful.
Lines 70-72 in the new version - We added sentence and reference on biological control.
Results
According to Figure 1, Prochloraz was also among the best fungicides controlling the fungal growth. Is there a specific reason why the authors did not include this fungicide in the field trial as well?
For field trials, we selected two of the best fungicides, each representative of a different mode of action. Prothioconazole and Prochloraz were both DMIs, therefore we used in the field only Prothioconazole.
Line 115, what do you mean by “very moderately” ? please replaced it with an appropriate adjective.
Line 121 - We modified the sentence.
Line 135, please replace the word “blocked” with an appropriate verb
Line 143 - Replaced with “inhibited”
Line 138, please rewrite again
Lines 146-148 - Rewritten
Line 140 , please rewrite again
Lines 149-152 - Rewritten
Line 148, not correct English,,, please rewrite
Lines 158-163 - Rewritten
162 what do you mean by “positively inhibited”. Please replace it with another word.
Line 176 - We modified the sentence.
Line 172 replace the word “block”
Line 187 - Replaced with “inhibited”
Figure 4, why did you combined all the fusarium together ? It will be better if the authors can split the Fusarium species rather than combining all the species.
The trial to which Figure 4 refers was aimed to evaluate Aspergillus flavus response to fungicides, by inoculating A. flavus strains on maize plants. We reported Fusarium data as natural contamination detected in the field.
Since the Fusarium contamination detected on the plants of this trial (focused on Aspergillus) was considered as natural contamination, the split at Fusarium species level was not our focus in this part of the experiment.
Discussion
Since the authors didn’t investigate the mycotoxins, therefore please focus only on the fungus not the toxins because the production of the mycotoxins (either suppression or enhancement is unknown).
The discussion is too long and needs to be shortened to avoid the confusion may happen to the readers.
We have limited the comments on mycotoxins.
We reduced and improved the discussion, trying however to present our results compared to literature papers for each fungicide, discussing also the potentiality and the disadvantages of the tested fungicides in terms of known resistance. Moreover we highlighted the crops for which these fungicides are currently registered or used.
Materials and methods
Line 446, “probably never or rarely exposed”, what do you mean ? either it is exposed or not ! please make sure about the information and restate it.
We removed the sentence.
Line 447, the concentration of the PDA is the same concentration stated by the manufacturer. Please delete it as long as there is not difference or replaced it by “according to manufacturer’s instruction”
Line 433 - We deleted the PDA concentration.
Line 448, how did you grow the fungi ? in Petri dishes ? which size.. 90 mm ? please state.
Line 432 - We added information.
Table 1, Please provide the references “if any” for the information you state ?
We added references as notes below the Table 1.
Line 463, did you autoclave the fungicides ? what about their stability at such a high temperature? If not then please write the sentence again.
Line 453 - Obviously fungicides have not been autoclaved, but they have been added after the sterilization of the PDA.We modified the sentence.
Line 478 how did you measure the diameter ? by a ruler or a digital calibre ? please state either in cm or mm
Lines 469-471 - We added information.
Line 494, what kind of microscope did you used ? please state
Line 486 - We added information.
Line 523, what media did you use to store the conidia ? sterile water ? or PBS? Please state
Line 513 - We added information.

Round 2
Reviewer 2 Report
the authors have responded to all the comments. no further corrections are needed.